# Negligible-cost microfluidic device fabrication using 3D-printed interconnecting channel scaffolds

Harry Felton[1]*, Robert Hughes[1]*, Andrea Diaz-Gaxiola[2,3]*

1 Faculty of Engineering, Mechanical Engineering, CAME School, University of Bristol, Bristol, United Kingdom, 2 Faculty of Engineering, Aerospace Engineering, CAME School, University of Bristol, Bristol, United Kingdom, 3 Faculty of Science, Bristol Centre for Functional Nanomaterials, School of Physics, University of Bristol, Bristol, United Kingdom

☯ These authors contributed equally to this work.
* harry.felton@bristol.ac.uk (HF); robert.hughes@bristol.ac.uk (RH); andrea.diazgaxiola@bristol.ac.uk (ADG)

## Abstract

This paper reports a novel, negligible-cost and open-source process for the rapid prototyping of complex microfluidic devices in polydimethylsiloxane (PDMS) using 3D-printed interconnecting microchannel scaffolds. These single-extrusion scaffolds are designed with interconnecting ends and used to quickly configure complex microfluidic systems before being embedded in PDMS to produce an imprint of the microfluidic configuration. The scaffolds are printed using common Material Extrusion (MEX) 3D printers and the limits, cost & reliability of the process are evaluated. The limits of standard MEX 3D-printing with off-the-shelf printer modifications is shown to achieve a minimum channel cross-section of 100×100 μm. The paper also lays out a protocol for the rapid fabrication of low-cost microfluidic channel moulds from the thermoplastic 3D-printed scaffolds, allowing the manufacture of customisable microfluidic systems without specialist equipment. The morphology of the resulting PDMS microchannels fabricated with the method are characterised and, when applied directly to glass, without plasma surface treatment, are shown to efficiently operate within the typical working pressures of commercial microfluidic devices. The technique is further validated through the demonstration of 2 common microfluidic devices; a fluid-mixer demonstrating the effective interconnecting scaffold design, and a microsphere droplet generator. The minimal cost of manufacture means that a 5000-piece physical library of mix-and-match channel scaffolds (100 μm scale) can be printed for ~$0.50 and made available to researchers and educators who lack access to appropriate technology. This simple yet innovative approach dramatically lowers the threshold for research and education into microfluidics and will make possible the rapid prototyping of point-of-care lab-on-a-chip diagnostic technology that is truly affordable the world over.

**Data Availability Statement:** The data is available at the following DOI: 10.5523/bris.34ac9q8m7ulgl2msiaiyfbv3zu.

**Funding:** Dr Hughes would like to acknowledge the training & support of Dr Annela Seddon and

BristolBridge (grant number EP/ M027546/1) under the Engineering and Physical Sciences Research Council (EPSRC - https://epsrc.ukri.org/) Bridging the Gaps between the Engineering and Physical Sciences and Antimicrobial Resistance cross-council AMR initiative, as well as pump-priming funding awarded by the Civil, Aerospace & Mechanical Engineering (CAME) School, University of Bristol. Harry Felton's work was undertaken as part of the Twinning of Digital-Physical Models During Prototyping project, funded by the EPSRC [grant number EP/R032696/1]. Andrea Diaz Gaxiola's work has been funded by CONACYT, MEXICO (http://www.conacyt.gob.mx/). Those individuals & funders acknowledged had no role in study design, data collection and analysis, decision to publish, or preparation of the manuscript.

**Competing interests:** The authors have declared that no competing interests exist.

## 1. Introduction

Over the past few decades lab-on-a-chip (LOC) technology has been heralded as the answer to a range of biological, chemical and global-healthcare challenges [1–4]. While most of the work done to date has been in research laboratories, some of the most exciting and impactful applications for LOC technologies are in the development of rapid point-of-care (POC) diagnostics for infectious diseases in developing countries, where resources for healthcare are most scarce [5]. Potential applications include the isolation and detection of non-communicable diseases in informal settlements and remote rural areas in lower middle-income countries (LMICs). However, in-spite of the decades of research invested already, LOC technology is yet to see meaningful adoption, research, and deployment in the LMICs where it is often of most value. The reason for this is most likely a question of cost, both at the research level as well as the mass manufacturing stage.

LOC technologies are underpinned by the field of microfluidics, referring to the control and manipulation of fluid volumes of the order μL or nL. This takes place within micro-scale channels where macroscale behaviour breaks down and unique microscale phenomena dominate [6]. The precise manipulation and analysis of chemical and cellular behaviour, often performed under microscope, is then achievable.

The most common method of microfluidic channel design and fabrication is the photolithography masking of a channel negative onto a crystalline Si wafer, producing a master mould of the microfluidic system. This master is then used as a mould for a biocompatible polymer such as polydimethylsiloxane (PDMS) which is poured and cured over the mould before being peeled free and bound to glass or aligned against another PDMS layer. This widely used multi-step procedure is well documented in literature such as [7]. However, photolithographic techniques and equipment are expensive and time-consuming. Their expense is further compounded by the iterative nature of the design and optimisation process when developing new microfluidic systems. This makes the widespread research and development of LOC technology near impossible for all but the wealthiest of research institutes and industries [8].

Due to the relative inaccessibility of these advanced fabrication methods, low-cost microfluidic and LOC technology has attracted considerable interest from many authors over the past decade. These have included the use of 3D-printed structures [9–15], milled glass and acrylic [16,17] and xurography (patterning with a cutter) [18,19]. Others have used paper microfluidics for disposable tests [20–23] and demonstrated the use of water-soluble 3D-printed approaches [5,24–26]. Despite the efforts of researchers, none of the methods yet proposed have answered all the challenges set forth, nor been widely adopted by the wider community.

N. Bhattacharjee et al. completed a study in 2016 that discussed the limitation of 3D printing for microfluidic channels [14]. For Material Extrusion (MEX) printed (referred to as Fused Deposition Modelling (FDM) in their work) microfluidic channels, three challenges were raised:

1. The printed parts could not be arbitrarily joined at the channel intersections.

2. Weak seals often occur between layers due to poor structural integrity.

3. And the size of the extruded material is often larger than the typical channel sizes used in microfluidics.

Recent work has tried to overcome these issues. Improvements in the accuracy and affordability of 3D printing technologies have presented a potential alternative to photolithographic

master fabrication methods. This has allowed researchers to overcome issue three above. Other research has explored 3D-printing entire microfluidic systems [27,28], as well as producing 3D-printed PDMS masters using inkjet, stereolithography (SLA) and MEX methods–which allows the first issue to be overcome. These processes typically demonstrate resolutions of 40–80 μm for inkjet printing [12], ~50 μm for SLA and 200–500 μm for MEX [29]. Although photolithographic techniques achieve channel resolutions of 10–15 μm [7], the resolutions achieved by modern MEX printers are now capable of printing at microfluidic scales.

Aside from the limitations around the resolution of 3D printing methods, one of the main issues with 3D-printing PDMS master moulds is the roughness of the surfaces. This roughness is caused by deposition occurring in the finite strata in 2 axes. A property of 3D printed moulds, also shown in work by Dahlberg et al. [26], are the ridges that can be observed in subsequent PDMS channels (shown in Fig 1E–1G). This results in poor adhesion to glass or PDMS substrates after curing and can result in a high failure rate during manufacture and testing. In addition, the limits of 3D-printing feature resolution mean that the edges of the channel form a gradual transition to the "bonding layer" that results in areas of low fluid flow where particles & fluid stagnate, becoming permanently stuck (Fig 1B and 1C). This leads to contamination and fouling of the channels, preventing their reuse. The gradual transition could also lead to high stress concentrations under pressure and may increase the likelihood of channel rupture during testing. Another drawback of using traditional 3D-printed master moulds is their degradation under the thermal cycling used to cure the PDMS. This can lead to warping of the substrate layers over time, limiting their lifetime of use.

The need for a low-cost, highly-accessible fabrication technique is discussed thoroughly by D. Rackus et al. [30]. The work highlights that there is insufficient literature surrounding the teaching of, and with, microfluidic devices in undergraduate teaching environments. The paper also recognises that there is a requirement for processes intended for earlier stage

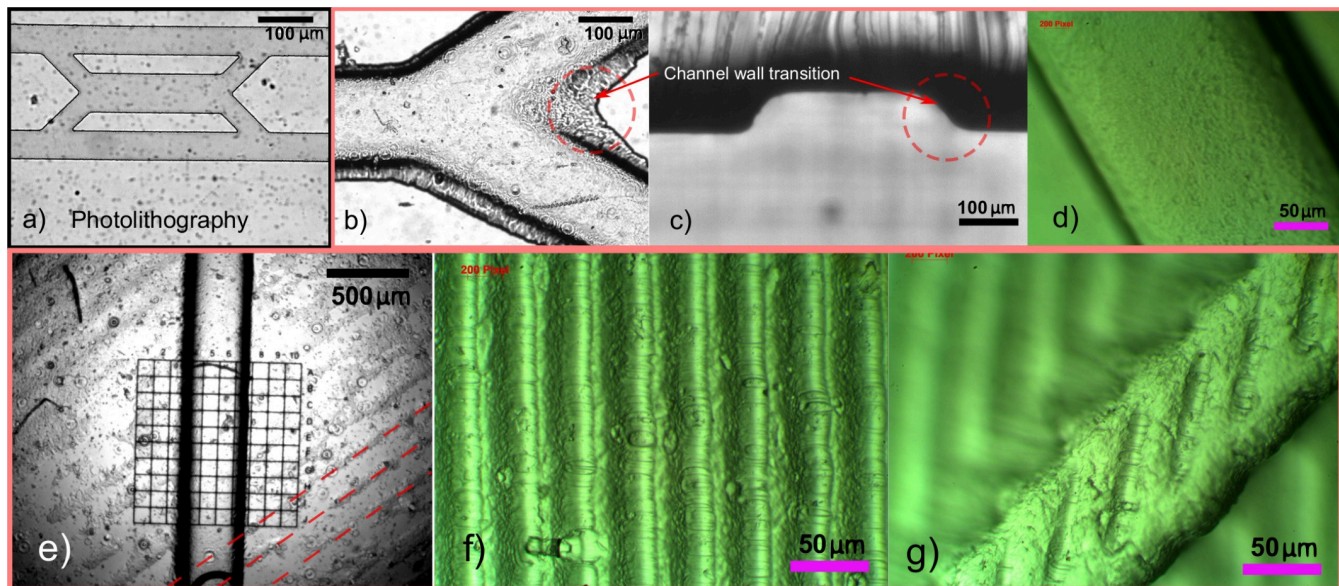

**Fig 1. Microfluidic master mould techniques showing microscope images of PDMS microfluidic channels manufactured using master moulds fabricated in different ways.** a) photolithography, and b-f) stereolithographic (SLA) 3D printing. Images b-d show channels manufactured using SLA moulds printed in-plane with the substrate to achieve smooth surfaces. Images b & c highlight (red-dotted circle) the gradual channel wall transition of the method. Images e-g show characteristic ridges of channels manufactured using SLA moulds printed at an angle to the substrate plane to improve print resolution.

education (up-to undergraduate level) and "outreach" events that those working in Science, Technology, Engineering and Mathematics (STEM) industries will have become familiar with. Several fabrication methods are discussed and their suitability for these applications considered, but no single method is recognised as a solution to the problems that need be overcome.

This paper presents a novel low-cost technique for rapidly manufacturing cheap microfluidic channel master moulds, employing MEX 3D printing and a simple interconnecting (click-and-connect) module design, thereby enabling low-cost microfluidic device fabrication. Through these innovations, the author's believe they have made important progress toward overcoming the challenges set out by Bhattacharjee et al. [14]. Details of the techniques and resources required are provided along with an investigation into the physical limits of the manufacturing methods and example microfluidic systems. The simple fabrication procedure can be accomplished using simple domestic equipment and a standard desktop, commercially available, 3D printer. This approach reduces the cost and complexity of microfluidic fabrication, making it universally accessible for research as well as education. In addition, all the fabrication processes have been developed in free-to-use software that is intended for non-expert users. By increasing accessibility, it not only lowers the bar of entry to help advance the prototyping and development of low-cost LOC technology, but could also be used in education to inspire the next generation of LOC researchers.

## 2. Method and materials

The process for rapidly fabricating low-cost microfluidic device scaffolds is shown in Fig 2 and detailed in the following sections. It can be broadly separated into 2 simple stages: The first, outlined in Section 3.1 involves 3D-printing, on mass, interconnecting microchannel scaffolds (or modules) directly onto a build plate (without platform, raft or skirt substrate). After their removal from the printer, the second stage, outlined in section 3, involves thermally bonding these microchannel modules to a glass substrate in the desired configuration–assembled using the interconnecting ball-and-socket joints (see Section 2.1)–to produce a microfluidic device master mould. These masters can then be used, and reused, to produce microfluidic devices in PDMS. Post-printing, the master mould fabrication process (Fig 3 steps 4–7) can be completed in under 5 minutes, allowing the method to be used for both informal and formal learning environments, using the criteria proposed by Rackus *et al.* [30]. In this section the design and manufacturing considerations of the first stage are presented and discussed.

### 2.1. Interconnecting microchannel design

Rectangular cross-section microchannels were designed in Computer Aided Design (CAD) (Autodesk Fusion) to be printed by a MEX printer directly on the build plate. Designs were created in a range of modular patterns with each featuring interlocking ball-socket connector end designs shown in Fig 3A and 3B. The inter-locking ends of the modules were developed to mimic puzzle-pieces such that successive modules could be arranged in any configuration desired. This allows easy integration and fabrication of multiple modules to produce more sophisticated microfluidic systems using a small number of simple modules. The connector geometry was designed to account for the thermal and physical phenomena of the MEX process. For example, the ball-connector is under-sized to account for thermal-spread—the effect of the material spreading slightly on the build plate to create wider-than-desired extruded lines. This effect, for 0.4 mm nozzle extrusions, results in extruded paths exceeding design dimensions by 0.1 mm [31]. Similarly, the tapered hole of the socket-connector counteracts the effect of thermal spread to allow the ball-connector to be positioned in a planar configuration within the socket-connector. The connector geometry, once the scaffolds have cooled, is

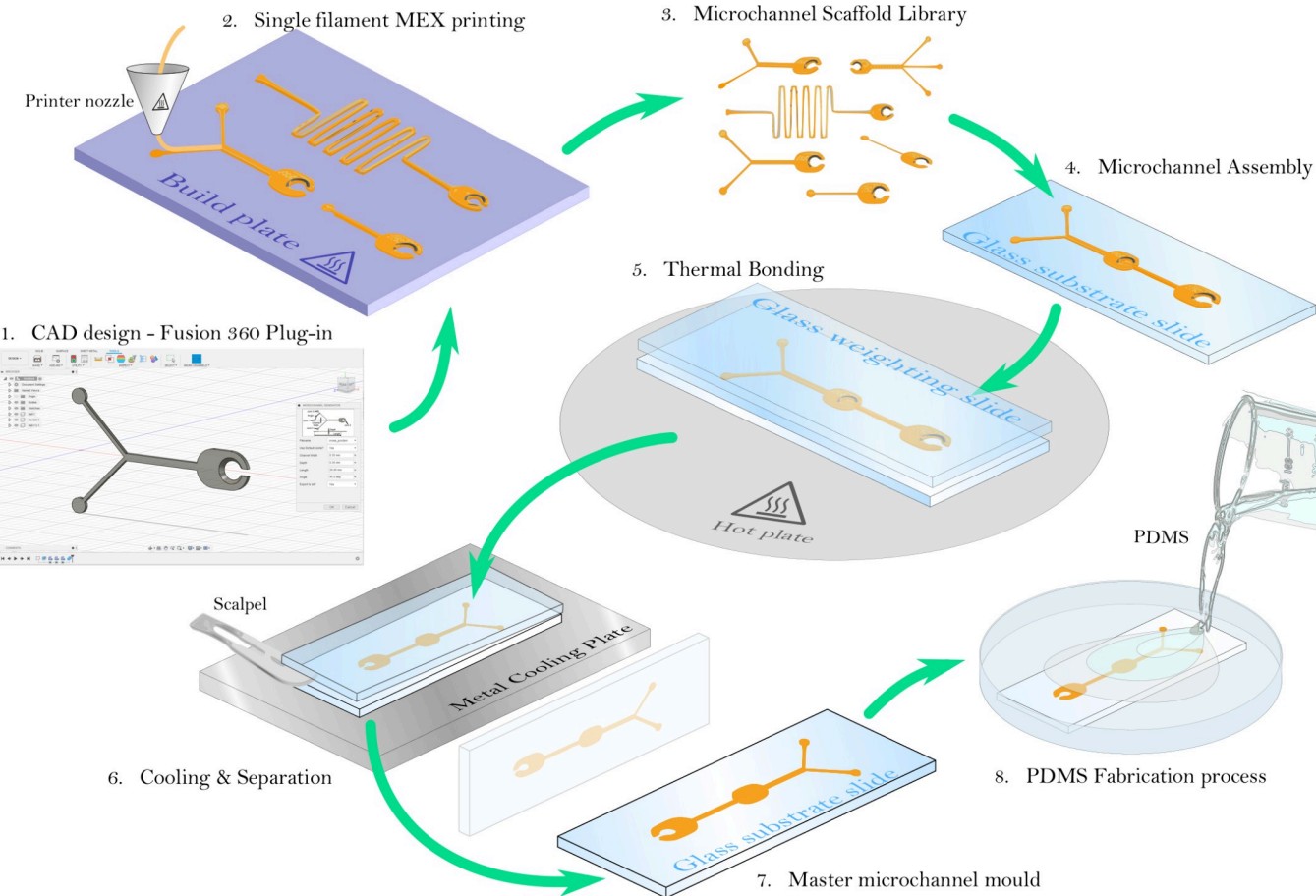

**Fig 2. Proposed microfluidic master mould fabrication process.** 1. CAD design of microchannel module using the open-source Fusion 360 plug-in, 2. 3D printing of microchannels scaffolds, 3. Library of pre-printed channel scaffolds, 4. Assembly of interconnecting channels on glass substrate, 5. Thermal bonding of channels to substrate, 6. Rapid cooling on a metal plate to create asymmetric thermal dissipation allowing the weighting slide to be easily removed without peeling the mould from the substrate slide, 7–8. The master mould can then be used in the PDMS microfluidic process.

at the capability limit of the MEX process using current technology and smaller connector designs were unsuccessful.

Within the scope of this study, five microchannel modules were explored: cross-junction, droplet generator, fluidic resistor, straight channel, and Y-junction. These are shown as CAD renderings in Fig 3 with their default joint configurations (annotated diagrams of each channel are available in S1 Fig highlighting modifiable features). These module designs were used throughout the proceeding experiments to test and evaluate the interconnectivity and suitability of the microchannel fabrication process (see Section 3).

## 2.2. Printing microchannel scaffolds

MEX, also commonly referred to as Fused Deposition Modelling (FDM) and Fused Filament Fabrication (FFF), has become a mainstream prototyping and production method for a range of engineering components. Ultimaker, one of the leading machine manufacturers in the sector, have several demonstrable success stories where MEX has been applied to a complex manufacturing problem [32]. This is supported by Sculpteo, who have found that the MEX method of manufacture has led the 3D printing market for several years [33–36] and is often used for production purposes [36].

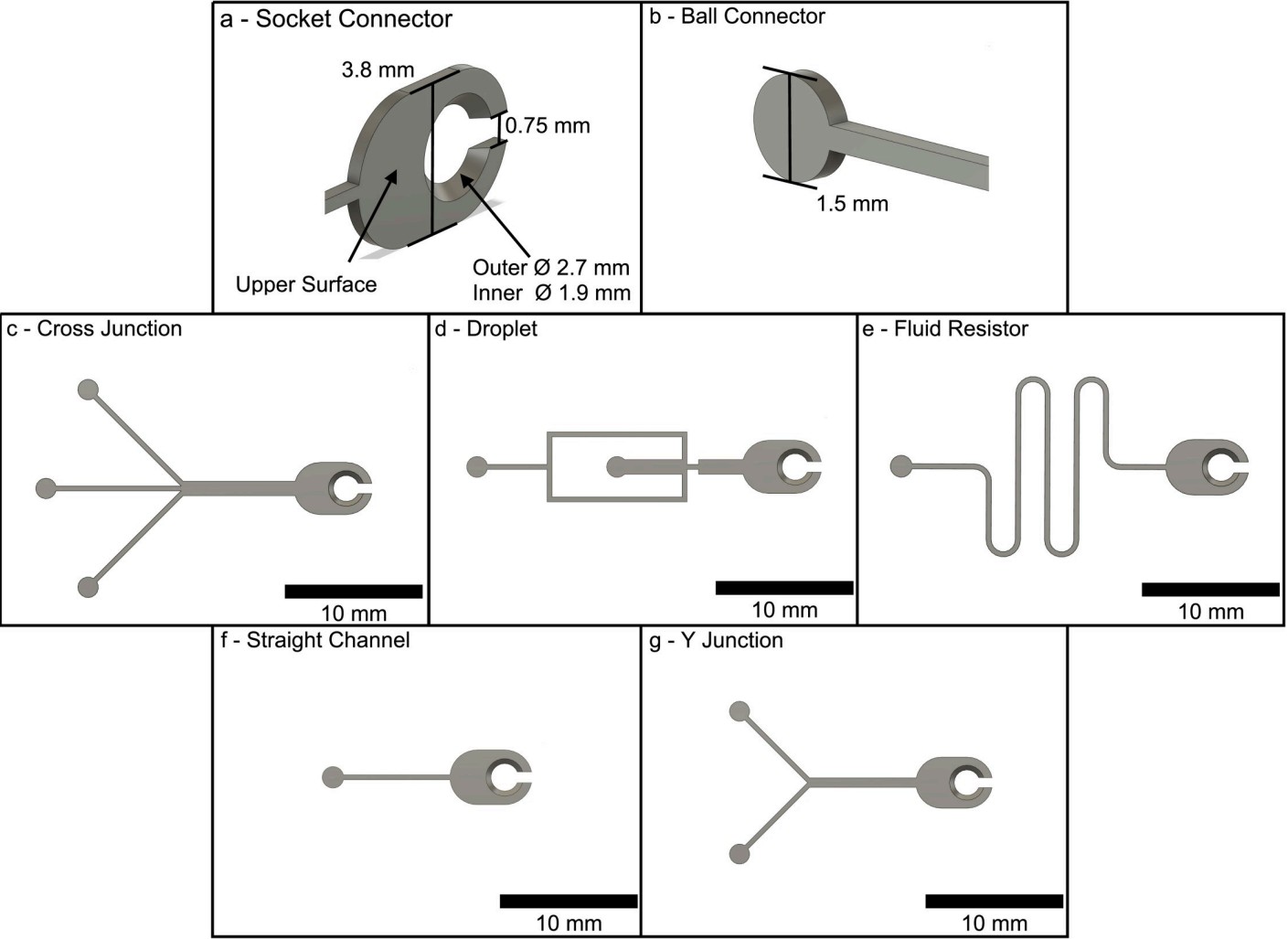

**Fig 3. Connector and module designs.** (a) & (b) show the CAD models for the inter-locking socket & ball connectors, respectively. (c)-(g) show the planar microchannel designs used in this study: (c) Cross-junction, (d) Droplet Generator, (e) Fluid Resistor, (f) Straight Channel, (g) Y-junction.

Improvements in the resolution and accuracies achievable from desktop MEX machines have helped drive the technology's adoption, coupled with reductions in the cost of machine acquisition and ownership. A Prusa Mini, for example, is currently available for GBP £324.26 (USD $408.57) [37]. As well as this, MEX parts require little to no post-processing when compared to SLA (or similar) technologies at a similar price point. Thus, the overall barriers to entry are low and outputs well refined.

The work detailed here, is intended to be clear and easy to replicate for the non-expert user. As such, more advanced techniques such as adaptive print path planning have been avoided and the process simplified as far as was reasonably practicable–for the example of print path planning, the automated generation process undertaken by Cura has not been modified. Quantitative analysis of the effect of changing print settings have also not been considered in detail within this paper, as this is an introduction to the fabrication method. Future work may consider how settings will differ between machines to achieve comparable results.

**2.2.1. Available equipment.** Throughout this work a single Ultimaker 3 Extended machine was used with Polylactic Acid (PLA) material. The machine is a standard

commercially available unit and had not been modified by the authors, with the exceptions of an enclosure to control particulate exposure, PrintBite build surface to improve print bed adhesion and a 3D Solex Hardcore AA print core to allow nozzles of varying sizes to be installed. A standard 0.4 mm ICE steel nozzle was used as the baseline. All additions are available to buy commercially.

PLA was chosen due to its low cost, high availability, and high printing reliability and safety [38], whilst providing suitable thermal properties for the mould fabrication process (see Fig 1) [39]. Ultimaker brand PLA was primarily used for the work, though other PLA brands were tested with the only changes necessary for similar function to the presented related to the print temperature used. An enclosure with High-Efficiency Particulate Air (HEPA) filter and fan was utilised to reduce the particulate emissions. Though this will have influenced the temperature profile around the print, the temperature of the print bed was expected to dominate this profile due to the small z-heights of the printed channels. This was confirmed by altering the time between prints (allowing the print chamber to heat and cool differently) and repeating prints. No meaningful change in success or part accuracy was recorded.

**2.2.2. Optimising printer settings.** Ultimaker's open-source slicing software, Cura (v4.5), was used throughout the work. The list of changed settings to allow the microfluid channels to print reliably is extensive. As such, the list of settings in their entirety are not contained herein but are available as supporting material (data in S2 Supporting information). Table 1 shows the most important changes to allow the channels, of minimum width 350 μm, to print reliably and with acceptable accuracy.

The printing of microfluidic scaffolds requires very fine and accurate material deposition, so the settings were modified to support this. For example, the line width was set to 200 μm to ensure the channels were printed as single line-extrusions whilst also improving the print reliability. The choice of settings also facilitated printing directly onto the build plate (using the PrintBite+ print surface on top of the standard glass build plate), without the use of a substrate. The absence of a substrate reduces the time and economic cost to print channel modules–by reducing the material that need be printed–and improves channel morphology and interconnectivity.

The print speed was found to have the most significant effect on the success of the microchannel prints as this improved channel adhesion, preventing peeling from the build plate. By lowering the print speed to 9 mm/s (the lowest speed achievable by the Ultimaker 3 Extended) this issue was overcome.

**2.2.3. Standard print quality.** In order to assess the standard print quality, several channel moulds were printed and analysed. For each of the channels shown in Section 2.1, 50 channels of width 350 μm were printed using the ICE 0.4 mm nozzle, giving a total of 250 channels. Of these channels, 96% were successfully removed from the build plate, with 4% failing during the printing process or were too badly damaged to remove from the build plate.

The channels successfully removed from the build plate were qualitatively assessed as useable or not. The definition of useable in this context was based on:

**Table 1. Print parameters for the 0.4 mm nozzle.**

| | | | |
|---|---|---|---|
| **Initial Layer Height** | 270 μm | **Line Width** | 200 μm |
| **Layer Height** | 100 μm | **Default Build Plate Temperature** | 60˚C |
| **Initial Layer Print Speed** | 9 mm/s | **Print Cooling** | Enabled |
| **Initial Layer Travel Speed** | 75 mm/s | **Print Thin Walls** | Enabled |
| **Number of Slower Layers** | 2 | **Build Plate Adhesion Type** | Skirt |

- Whether the channel had printed as a continuous path without gaps

- Whether the socket and ball connectors had printed with definition such that:

  ○. They could interlock with other channels,

  ○. They did not have any holes.

The mean channel width was found to be 0.60 mm for the useable channel scaffolds as shown by the solid blue distribution in Fig 4B.

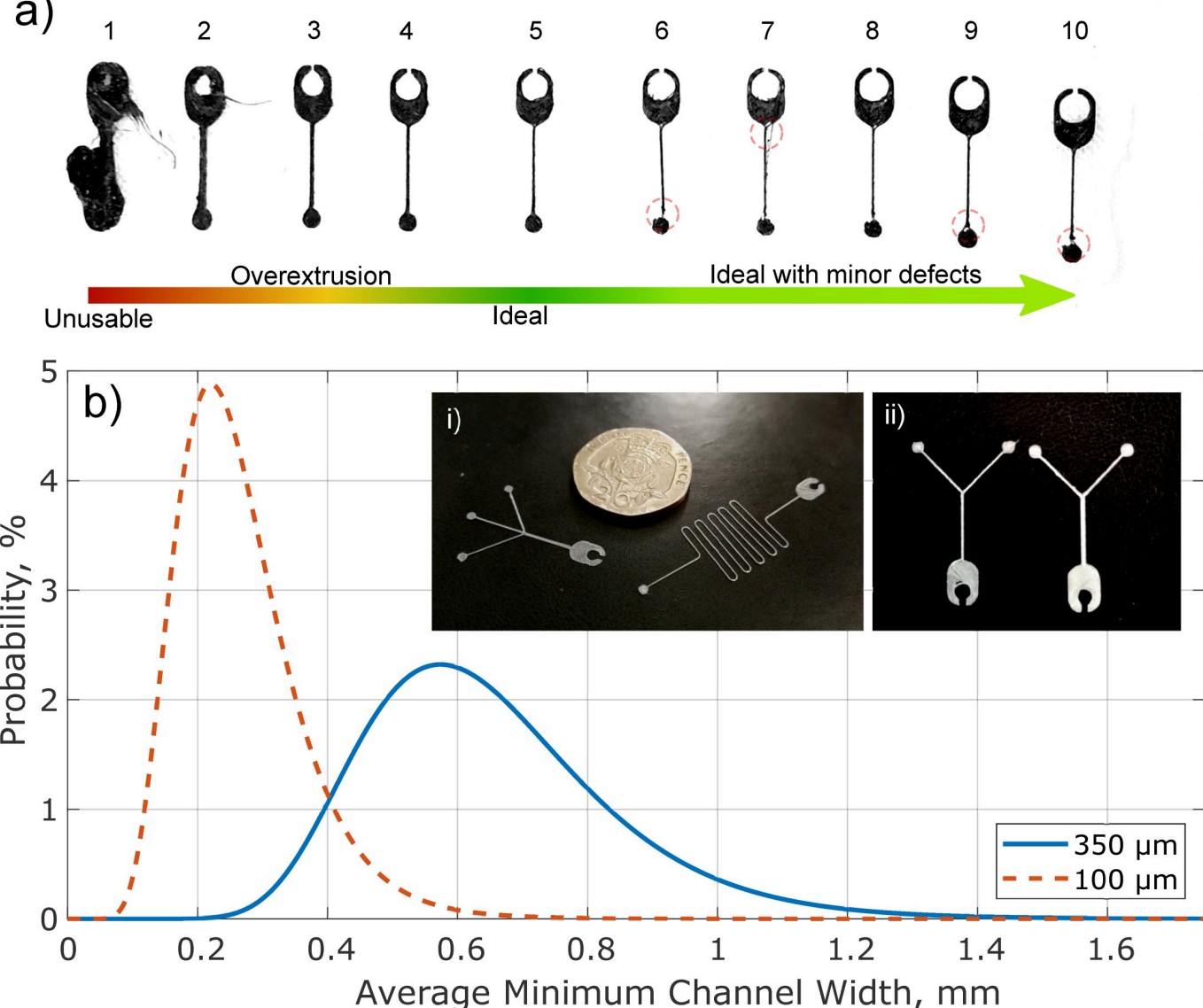

**Fig 4. Printed channel examples and distributions.** (a) Printing variation in straight channels, demonstrating the changes in extrusion due to poor pressure control, indicating unusable channels, over extruded, moderately extruded, and ideally extruded channels as well as highlighting (red dotted circles) minor print defects due to under extrusion. The channels were printed in order, from left (over-extruded) to right (under-extruded). (b) shows log-normal distributions for a set of 250 printed 350 μm and 100 μm channels with i) showing a cross-junction and fluid-resistor relative to a UK 20p coin and ii) comparing an ideally extruded Y-Junction channel (left) to a moderately over-extruded channel (right).

The print issues that led to printed channel scaffolds either failing or being unusable were traced back to 3 main issues; peeling when printing the ball-connectors, under-extrusion and over-extrusion–with all of these being significantly affected by the print bed levelling and filament control of the printer. For the 250 channels printed, 24 failed due to under-extrusion, 44 failed due to over-extrusion and 7 failed due to peeling of the ball-connector.

The time and cost of printing the microchannels was calculated using the CAD material volume and Cura time estimates. These estimates indicate that a 5000-piece library of microchannel modules, printed using the 0.4 mm nozzle, could be manufactured for less than USD$1.50, while the overall printing time for 10 channels varied between 7 (straight channel) and 27 (fluidic resistor) minutes with a mean print time to produce 10 channels across all designs of 15 minutes.

**2.2.4. Exploring printer limits.**   To approach the resolutions required for many microfluidic applications, the printing limits of the MEX technique were explored. To achieve this, a 3D Solex Hardcore 0.1 mm nozzle was installed—the smallest commercially available nozzle that could be found for the Ultimaker 3 Extended printer. Though a sub-0.1 mm nozzle could be made, it was important to this study that the equipment used was limited to what was commercially available. This was done to demonstrate a method that was accessible to the wider educational, health care and academic sectors. The total cost of modification was ~£242 (for the new print core and nozzle).

The channel designs detailed in Section 2.1 were tested with the 0.1 mm nozzle with a designed channel depth and width of 100 μm. Only two settings were changed between the 0.4-mm and 0.1-mm nozzle within Ultimaker Cura (though several related settings were automatically updated). These were reducing the initial layer height to 75 μm and reducing the line width to 90 μm.

By monitoring the print, it was possible to improve the pressure control by feeding the material by hand when the printer started to under-extrude. Though this increased complexity and heighted the failure rate it was possible to print several channels in succession that were suitable for use. Fig 4B shows the distribution a print of 250 channel scaffolds produced, with 167 successfully printed channels (66.8%) and 156 useable channels (93.4% of successfully printed channels)–defined using the same criteria as in Section 2.2.3. There was significant variability in the print success of the 100 μm channels, with much of this attributed to the accuracy of the print bed levelling and material feed rate. As technology moves forward, improvements to feeder control will come as standard on desktop printers; for example, the Ultimaker S5 already incorporates dual-geared feeders [40] and it is hoped similar improvements will be made in the way print bed levelling is undertaken.

As the volume of material extruded to manufacture the 100 μm channels was less than the 350 μm channels, the material cost was lower (even after the failure rate was considered). A 5000-piece module library (1000 of each module as discussed for the 350 μm modules) would cost ~USD$0.50. As such the material cost of a 100 μm channel mould is effectively negligible.

The print time increased compared to the 350 μm modules with an average production time for 10 modules of 23 minutes–an increase of over 50%. This was primarily because the connector designs were kept consistent, and so the nozzle had to physically move further to extrude the same amount of material from the smaller nozzle. It may be possible to change the connector design to improve the print times and further reduce the material costs, but this has been left for future work.

## 2.3. Open-source microfluidics

To ensure that the proposed technique is fully democratised, an open-source Autodesk Fusion add-in has been developed (MicroChannels.py in S1 Supporting information), allowing any

user to design and export interconnecting microfluidic channel scaffolds for 3D printing. Through the use of this plugin and the method detailed above, a user can go from a microfluidic channel design, through to a completed microfluidic channel without requiring CAD software expertise or the use of any time or resource intensive techniques or equipment.

In addition, if a user does not have a suitable 3D-printing capability, a microchannel module library toolbox could be requested from the authors or other 3D-printing facility.

A protocols.io instruction set has been made available detailing the full process, with links to the up-to-date add-in and profiles, available here: https://dx.doi.org/10.17504/protocols.io. biw7kfhn [41].

## 3. Microfluidic mould fabrication

Before the printed channel designs can be used as microfluidic master moulds, they must first be bonded to a planar substrate. The smoother and cleaner the substrate surface, the better the binding will be of the resulting cured PDMS cast to glass or other PDMS layer. Hence the microchannel modules are fixed to a glass substrate via a simple thermal bonding process detailed below.

### 3.1. Thermal bonding to glass substrates

PLA compounds are known to start melting between 145 and 160˚C [42], though the material's viscosity only drops low enough to be printed (i.e. flow) between 190 and 210˚C. To achieve thermal bonding to a glass substrate, and join interconnecting modules to form a single microfluidic channel mould, the scaffolds must be heated to a temperature such that the scaffolds melt and fuse–preliminary investigations found this temperature to start at 180–190˚C. The following section details the optimised thermal-bonding process.

**3.1.1. Thermal bonding process.** The 3D-printed mircochannel modules are mounted onto standard 1 mm thick glass microscope slides (£2.10 per pack of 50—Code: MIC2000— from Scientific Laboratory Supplies, UK) into the desired configuration, using the ball-and-socket connectors. Channels should be heated for between 45-60s (depending on the channel depth) on a hot plate at 200˚C to bind them to the glass substrate slide and smooth out inconsistencies. A glass slide placed on top of the channel scaffolds (weighting slide–see Fig 2.5) prevents the channels from deforming, either via shrinking or warping out-of-plane of the substrate slide. This effect occurs when some parts of the module are not completely flush to the glass surface leading to localised areas of thermal expansion in the thermoplastic (see Fig 5A). The weighting slide also acts as a uniform surface to smooth out inconsistencies in the surface of the scaffold, leading to more transparent microfluidic channels (see Fig 5D) and consistent channel depth (see Fig 5C). After heating, the slides (substrate & weighting), now partially fused together by the thermoplastic channel scaffolds, are placed weighting-slide-down onto a metal plate for a few minutes (see Fig 2.6). The high thermal conductivity of the metal plate promotes rapid cooling of the weighting slide to facilitate its easy separation from the mould, while maintaining good thermal bonding of the scaffold to the substrate slide.

**3.1.2. Heating-time evaluation.** The optimum heating time required to achieve scaffold bonding and smoothing is unknown. A study was therefore conducted to examine the effect of heating time on channel geometry and quality to optimise this process. Scaffold channels designed with a channel width & height of 500x500 μm were used in the study. The thermal bonding process detailed above was repeated 3 times for each heating time interval (15-60s, in increments of 15±2 seconds) and the resulting master moulds used to fabricate channels in PDMS. Microscopy and image processing were performed to evaluate the resulting PDMS microfluidic channel features (height, width, perimeter length and opacity). Cross-sectional

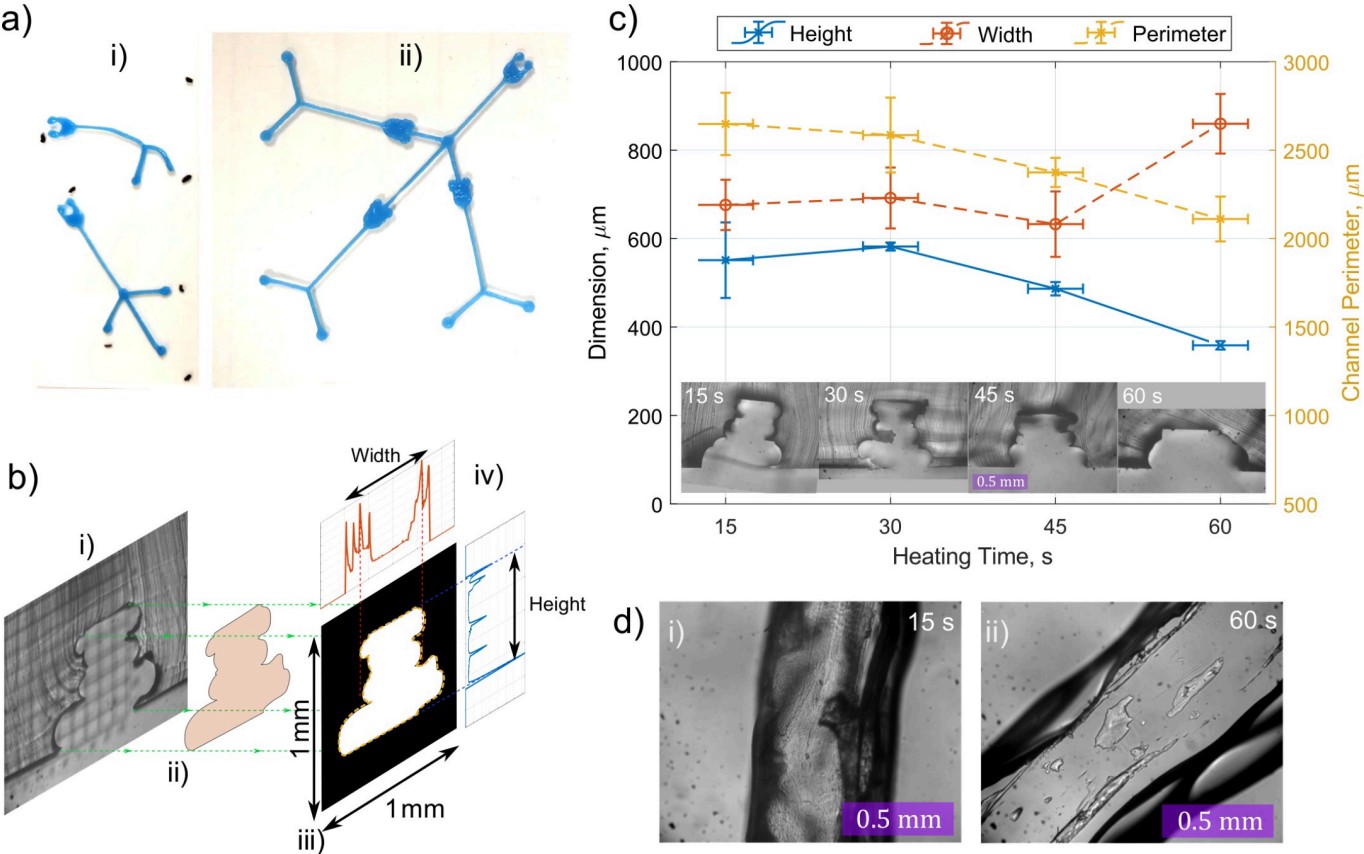

**Fig 5. Thermal bonding microchannel scaffolds to glass.** Image a) shows the deformative effects of the heating process on unweighted channel scaffolds (i), compared to those thermally bonded using a weighting slide (ii) to ensure thermal actuation does not occur. Diagram b) demonstrates the image-analysis process for analysing channel dimensions using segmentation from the raw microscope images (i-ii) to produce binary images for automated scaling analysis (iv). Graph c) plots the microfluidic channel cross-sectional dimensions and the channel perimeter as a function of heating time for channels printed at 500x500 μm (width x height). Error bars represent standard deviation from 3 repetitions. Microscope images d) show the top-down views of channels heated at 200˚C for 15s & 60s.

microscope images of the PDMS microfluidic channels underwent manual segmentation to extract the channel geometry for more accurate automated geometrical analysis (Fig 5B). The results of this study are shown in Fig 5C and 5D.

The results shown in Fig 5C indicate the change in aspect ratio of the channel as a function of heating time, along with the repeatability in channel dimensions between different prints. To assess the degree of irregularity in the channel cross-section, the perimeter length of the cross-section was measured. The greater the perimeter, the higher the degree of irregularity in the channel wall. It is clear from the Fig 5C that as the heating-time increases, the walls of the channel cross-section become more uniform and regular. Fig 5D shows top-down images of the channel surface for 15s and 60s heating times. The images show the reduction in surface roughness and opacity of the channels with heating time.

**3.1.3. Heating-time prediction.** A 2D finite element (FE) model was developed to simulate the transient thermal transfer process described above and was used to estimate the minimum heating time needed for the whole PLA scaffold to exceed the bonding temperature (180–190˚C). The FE model was developed using COMSOL Multiphysics 5.3 in the geometry shown in Fig 6 (t = 0s). The PLA was modelled as an acrylic plastic with a heat capacity of 1,800 J·Kg⁻¹K⁻¹, density 1.24 g·cm⁻³ and a thermal conductivity of 0.13 W·m⁻¹K⁻¹ [43]. The glass slides were modelled using the built in COMSOL silica glass material with a heat capacity

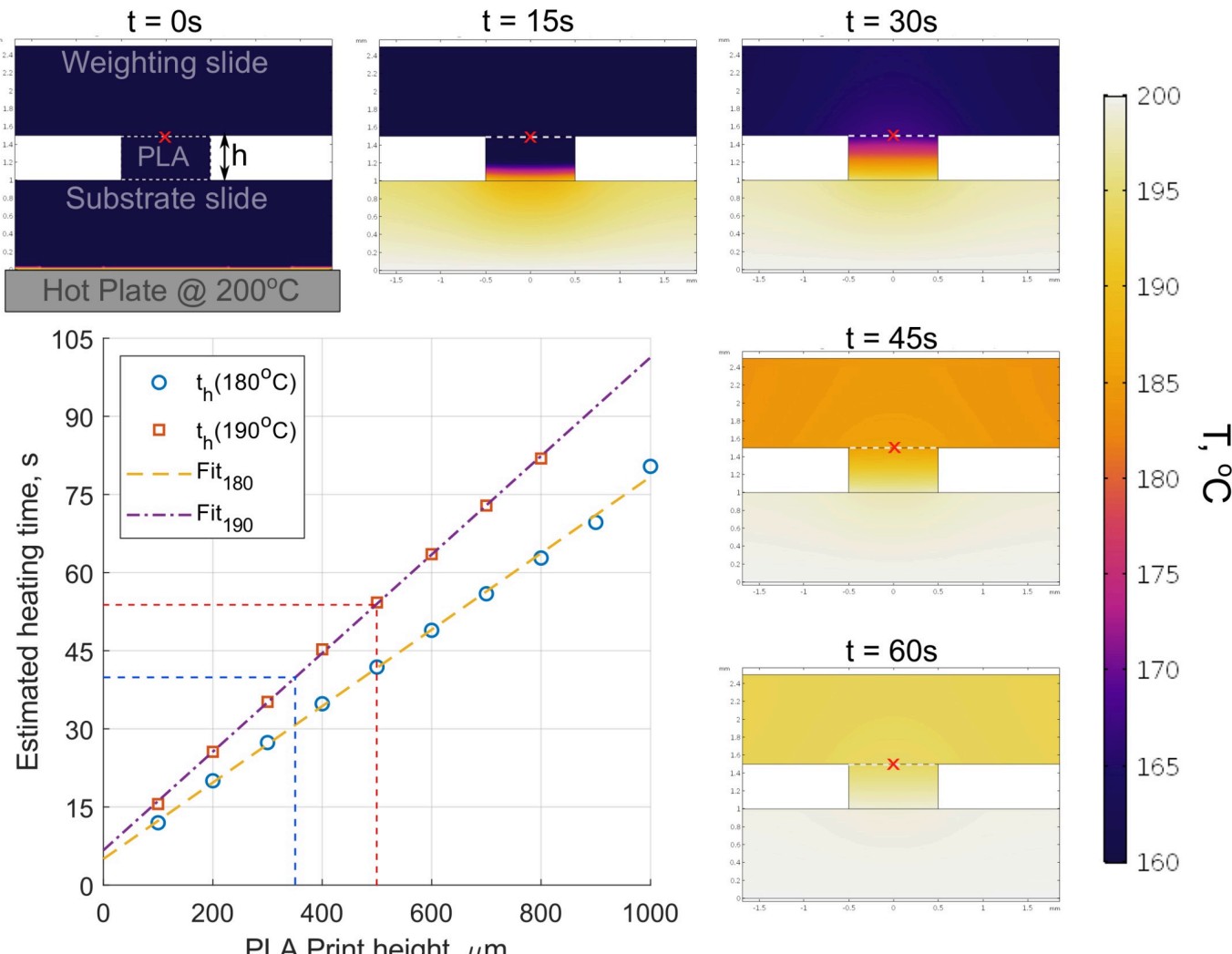

**Fig 6. 2D finite-element simulation of the thermal bonding process.** 2D finite-element simulation of the thermal bonding process, showing simulated temperature profiles at discrete time intervals (0-60s) of a 500 μm high PLA component between glass slides being heated from below. The graph shows the estimated time it takes for a test point at the top of a PLA component of height, h, to reach the bonding temperature (180˚C & 190˚C). The test point is identified in the temperature maps by a red cross.

of 703 J·Kg$^{-1}$K$^{-1}$, density 2203 Kg·m$^{-3}$ and a thermal conductivity of 1.38 W·m$^{-1}$K$^{-1}$. The results of the simulation are shown in the graph in Fig 6.

The time it takes for the entire PLA component to exceed the documented melting temperature (180–190˚C) is evaluated as a function of component height, h, and the linear relationship is shown in the graph in Fig 6. The simulated results support the observations made experimentally of the changes in channel morphology as a function of heating time (Fig 5C and 5D), with a ~500 μm high component requiring a minimum of 45 seconds before thermal bonding and deformation is observed. As the simulation does not account for the thermal deformation of the PLA, or the uneven geometries of the as-printed (t = 0s) scaffolds, the predicted heating times for a given scaffold height should be considered a minimum threshold.

**3.1.4. Microchannel characterisation.** The consistency in channel cross-section throughout a module design was accessed along with the repeatability of the fabrication process for that design. This study was performed on a 10-turn fluid resistor channel design (Fig 4E) with

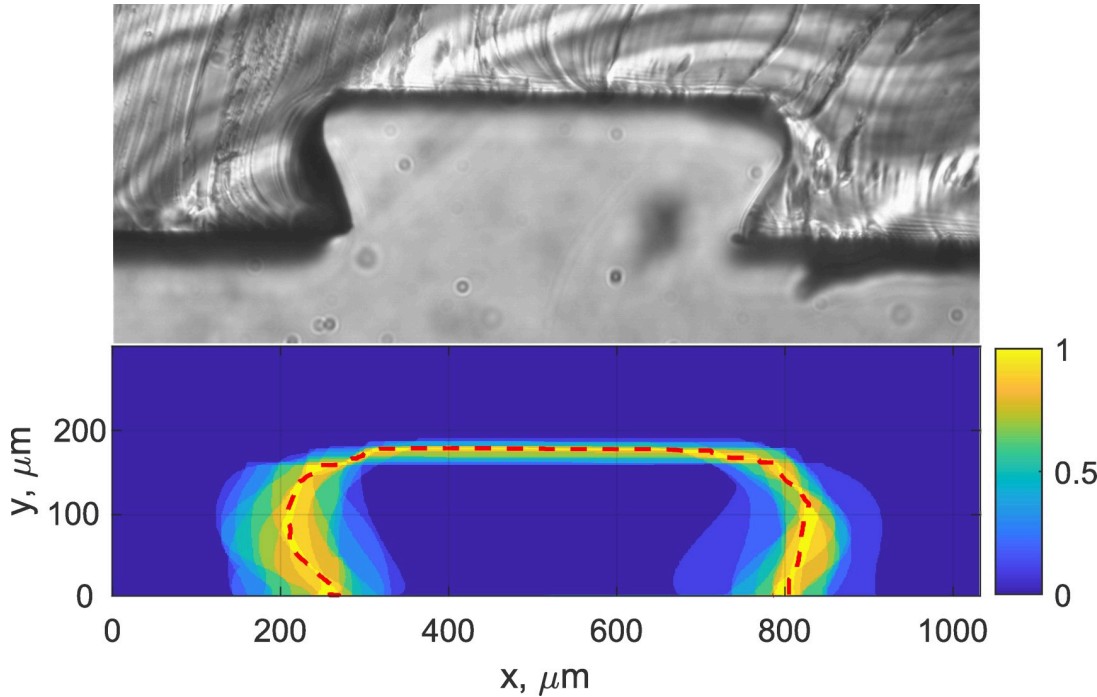

**Fig 7. Channel repeatability.** (Top) Cross-section of PDMS channel created using 3D-printed moulds (350x350 μm) heated for 50s at 200˚C, relative to (Bottom) a probability map of channel dimensions made from image analysis of 21 cross-sections measurements across 3 fabricated modules. The dotted line represents the mean channel edge.

cross-sectional dimensions of 350x350 μm in CAD. Three identical modules underwent the thermal bonding process (as detailed above) for 50±2s to ensure adequate deformation and bonding to the glass substrate, as informed by the FE model predictions shown in Fig 6. The resulting master moulds were used to fabricate microfluidic channels in PDMS, cross-sections taken and analysed from microscope images as summarised in Section 3.1.2 and Fig 5B. In total, 21 cross-sectional measurements were made across the 3 repetitions. The results are shown in Fig 7 and summarised in Table 2, with errors determined by the standard deviation about the mean.

It is clear from the results that the height of any given channel is consistent to within ±10 μm across the whole channel, while the width varies by up to ±100 μm. Equally, the thermal-bonding fabrication step exhibits acceptable repeatability between channels and produces favourable channel cross-sections with distinct transitions from channel wall to bonding layer. This property will reduce the potential for channel fouling, facilitating reuse, and will also reduce the concentration of corner stresses that could lead to rupture observed in other 3D-printed mould designs (see Section 1).

**Table 2. Channel repeatability of cross-sectional dimensions for 350x350 μm CAD-designed channels, heated for 50s at 200˚C.**

| Channel No. | Mean Height, μm | Mean Width, μm |
|:---:|:---:|:---:|
| 1 | 186 ± 3 | 540 ± 100 |
| 2 | 189 ± 5 | 630 ± 80 |
| 3 | 171 ± 7 | 700 ± 80 |
| Mean | 182 ± 5 | 630 ± 90 |

**3.1.5. Bonding to glass.** In most cases of microfluidic device fabrication, the cured PDMS channel castings are bonded to a substrate, typically glass or another PDMS layer. The most effective method of bonding uses a process of corona discharge, also known as plasma-activation, which introduces polar functional groups to the bonding surfaces of the PDMS and glass, allowing strong covalent bonding when brought together [44]. This process also acts to temporarily change the PDMS channels from hydrophobic to hydrophilic [45]. Although the process results in a near permanent bond, the equipment required to perform plasma-activation is prohibitively expensive for many. However, with the method presented here, one of the additional benefits is that the bonding surface of the cured PDMS cast has the same smooth texture as the glass substrate, allowing it to be applied directly to another glass substrate where it statically adheres. Something not possible when using moulds with 3D-printed substrates (Fig 1) due to the roughness and ridging effect discussed in Section 1. The efficiency of this binding was evaluated through a simple pressure test, performed under microscope.

Linear channel and Droplet generator chips were controlled with a Microfluidic Flow Control System (MFCS-EZ, Fluigent), which provides a pneumatic pressure feedback. Air flow from the pump is connected by luer-coupled 1/4 OD Tygon Tubing to a reservoir tube which in each case contains the substance to flow through the channel. These reservoirs are then connected to the chip through a 1/16 OD Tygon Tubing and a 23 G blunt needle.

For the linear chips, the reservoir was an empty Eppendorf tube, and air flow was provided in increments of 20 mbar, shown as black plots within in Fig 8B–8D, as *Input*, until the channels failed. The feedback pressure measured within the channel is shown as *Output*. Two bonding conditions were tested, defined as Plasma Bonded (PB) and No Surface treatment (N). In addition, the devices were fabricated with combinations of biopsy punctured Inlet (I) and Outlet (O) holes to accommodate inlet/outlet tubing. Given our proposed fabrication method, we used PB+I+O to provide a reference and compared their performance to N+I+O bonded chips.

From Fig 8, it can be noted that all chips exhibit compliant output pressures from 0–1200 mbar (17.4 psi), while N+I and PB+I+O reach above 1850 mbar (26.8 psi) without compromising the channel's integrity. Two types of failures have been identified, the first one, outlet failure, where the punctured PDMS inlet in contact with the needle suffers from deformation, and the air begin to escape through the biopsy puncture orifice. This is a non-critical failure since PDMS can recover from such deformation and allow it to be used for more than one testing cycle. The second type of failure is PDMS disbonding from the glass substrate. Pneumatic

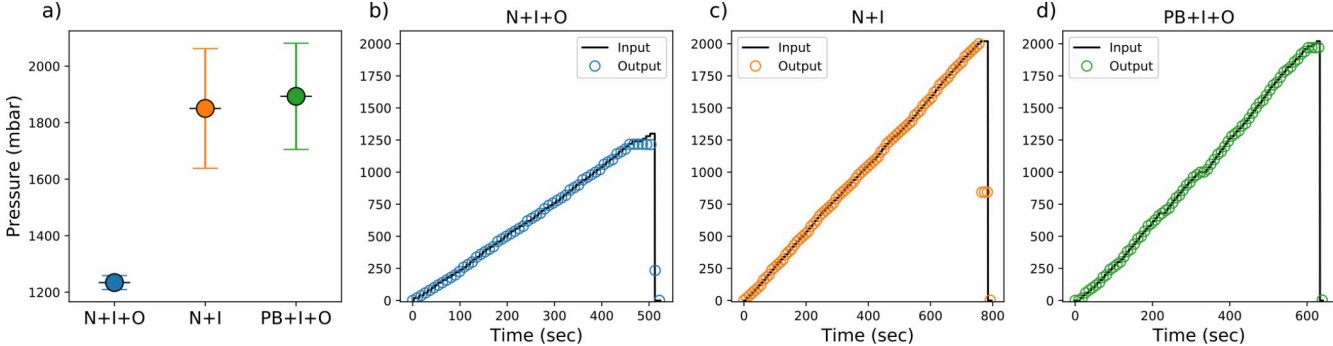

**Fig 8. Maximum pressure testing.** Groups of testing are labelled as Plasma Bonded (PB), No Surface treatment (N), Inlet (I), Outlet (O). a) Failure points for groups of tested linear channels, error bars are one STD. b) N+I+O behaviour, with inlet and outlet punctured into the substrate, c) N+I behaviour, only inlet punctured into the substrate and d) PB+I+O behaviour, with inlet and outlet punctured into the substrate.

pressure could drive the detachment mainly if the PDMS exhibits fabrication defects, for example, an uneven bonding surface.

PB+I+O and N+I+O groups are expected to exhibit non-critical failure since the outlet can behave as a regulatory valve. N+I group was expected to show PDMS detachment, due to the lack of an outlet. However, only one specimen within this last group showed evidence of debonding, the rest would show channel deformation in the inlet, holding the pressure steady after failure at 840 mbar (12.2 psi).

## 4. Microfluidic device demonstrations

To demonstrate the practical use of the fabrication technique and the interconnecting channel modules proposed herein, two microfluidic devices were manufactured and tested in PDMS: An interconnected dye-mixing device to demonstrate the flow between interconnected channels, and droplet generators at 350μm & 100μm. After completing the fabrication of each master mould (see Section 3), PDMS (Sylgard 184, Dow Corning, Midland, MI, USA) prepolymer was mixed in a 10:1 ratio with the curing agent, poured onto the master mould and degassed, before being cured for 24 hours at 40˚C (7). The cured PDMS was cut and removed from the master and a 0.4 mm biopsy puncture used to create inlet and outlet holes. The PDMS microchannels were then bonded to a glass substrate for use as microfluidic devices.

### 4.1. Dye-Mixing

A fluid mixing device was manufactured using the interconnecting module design. The device channel was formed by connecting a Y-junction and fluidic resistor module, as can be seen in Fig 9. The channels were manufactured from scaffold modules with a CAD-designed geometry of 350x350 μm and thermally bonded for 50s at 200˚C to produce microfluidic channel dimensions equal to those shown in Table 2.

Yellow and blue dyes were passed onto the Y-junction inlets (left, Fig 9C.i) using a continuous cycle dual syringe pump (KDS 270, KD Scientific Inc., Holliston, MA, USA) at a flow rate of 1000 μl/min. The dyes exhibit laminar flow when they meet at the Y-junction intersection as well as through the socket-ball joint. This is consistent with a low Reynolds number flow system in the Stokes-flow regime [46]. Within the fluid resistor channel, the dyes continue to exhibit laminar flow, although clear mixing between the dyes occurs resulting in a dilution of the blue dye into a green stream at the outlet.

### 4.2. Droplet generation

Droplet generator scaffolds were printed at 350μm and 100μm widths using the open-source Autodesk Inventor add-in (see Fig 3D) and used to fabricate devices in PDMS. Droplet generators rely on a T-junction where two identical transversal channels disrupt the flow of a perpendicular channel to achieve droplets with defined sizes. These microfluidic architectures have been widely explored since they are relevant to create coacervates and microspheres useful in different fields for microanalysis [47].

The droplet generator chips were demonstrated under microscope through the formation of alginate microspheres via internal gelation [48,49]. Alginate can be gelled in the presence of calcium ions that physically crosslink guluronic fragments. Insoluble $CaCO_3$ (60mM) is mixed in a neutral pH ungelled alginate 1.5%(wt.) aqueous solution. When in contact with acidic solutions, calcium carbonate dissociates into water, $CO_2$, and $Ca^{2+}$, triggering gelation. Here, the sheath oil phase, Mineral Oil, contains acetic acid (1:1000 vol. to that of min oil), allowing the flow time between the T-junction of the chip and the collection reservoir for it to diffuse into the sphere and crosslink it. This phase has been conditioned with a surfactant, Span 80

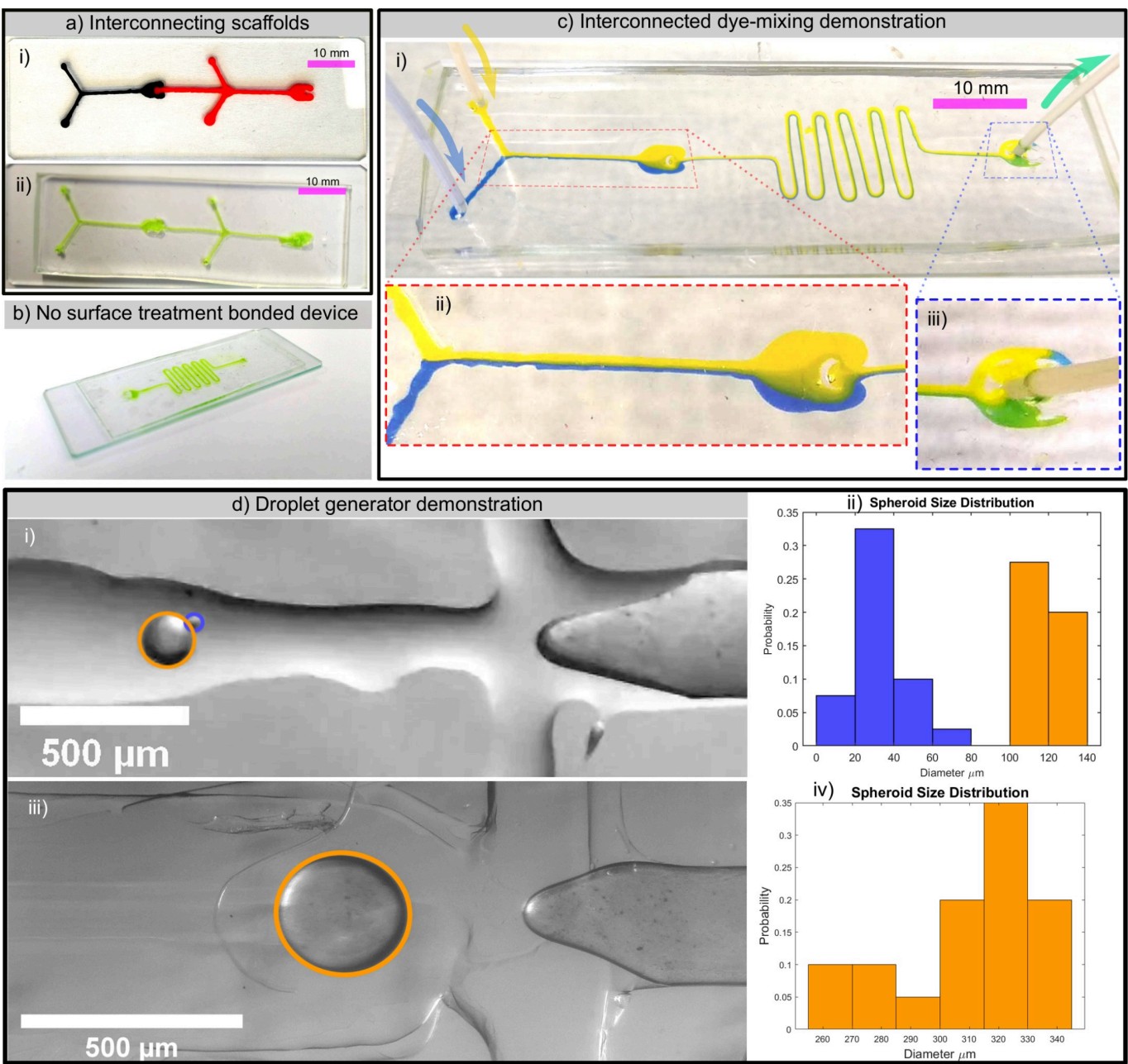

**Fig 9. Low-cost MEX 3D printed microfluidic device demonstrations.** a) thermally bonded socket-ball joint master (i) and subsequent PDMS microchannels on glass containing green fluid dye (ii), b) showing green fluid dye in a mixer microchannel in PDMS directly applied to glass after curing (no plasma bonding), c) interconnected 350μm microchannels in PDMS exhibiting laminar flow (ii) and dye mixing (iii), and d) demonstration of droplet generation showing droplets under microscope using 100 μm (i) and 350 μm (iii) scaffolds and the generated droplet size distributions using 100 μm (ii) and 350 μm (iv) scaffolds.

(1:10 wt. to min. oil) to stabilise the bead, avoid coalescence prior to gelation, and maintain the yield spheroid size distribution [50]. In each case, the pressure was adjusted to achieve single droplets. In both systems, alginate flow channel pressure was ≈10% (48 and 130 mbar respectively), to that of the oil phase (485 and 1280 mbar respectively), this can be taken as an indicator of the consistency of sizing within the fabrication process.

The droplet size distribution has been quantified for both 100 and 350 μm channels. These experiments have proven the correlation between channel geometry and droplet size. As has been discussed in the fabrication section, printed geometries are affected by a thermal expansion as a consequence to the thermal bonding process (see section 3.1.3). Each chip will be referred to as their original printing dimensions.

Sphere sizes produced with the 100 μm channel width yield a bimodal distribution (Fig 9D), within the 120±9 μm for the larger droplet size range, which correspond to those desired. Those occurring in the range of 34±11 μm correspond to a comet trail of the original bead. These smaller spheres could coalesce with the larger fragments if not in the presence of the surfactant. The size, shape, and rate of droplet formation for 100 μm channels can be observed as provided in S1–S3 Videos. Chips with a 350 μm channel yield droplets of 310 ± 24 μm in a single distribution. Both evaluated sizes demonstrate device's performance in the dripping domain, where the droplets demonstrate monodispersity associated to the channel's width.

This method for droplet generation was chosen for its relevance in biological assays since it allows to collect, measure and post-treat them. They are compatible with more complex systems for single and multiple cell encapsulation [51], cellular spheroid formations [50], or even wet-spinning approaches [52]. Which makes this fabrication method arguably the most economic path towards micro-biotesting.

Polydispersity Index (PDI) evaluates the width of the distribution of droplet sizes $PDI = s/d$, where $s$ is the standard deviation and $d$ is the mean of droplet diameter [53]. Literature droplet metrics report that analogue T-Junction droplet generating geometries yield spheres whose distribution can be as small as 1–3% [54]. PDI values lower than 10% are associated with lower incubation times, when these droplets are applied in biological assays. Non-microfluidic methods, usually high throughput ones, produce droplets whose PDIs range between 10–20%.

Linear channel testing (see Section 3.1.3) demonstrates the compliance of the herein proposed fabrication method for channels subject to pressures higher that conventionally used in microfluidic water or oil systems both commercially (700 mbar, 10 Psi) [55] or academic evaluations (860 mbar, 12.5 Psi) [56], or else, for highly viscous solutions. These regimes of pressure were also evaluated in MEX-fabricated droplet generator chips, which have demonstrated fidelity when it comes to final spheroid size, regardless of the thermal expansion of the channel width, as a consequence of the bonding process. These have also yielded droplet size distributions with PDI of 7.1% and 7.7% for 100 μm and 350 μm respectively. Although the designs have not been optimised to achieve lower polydispersity, this is a reasonable starting point if it is to be used as a teaching resource.

## 5. Discussion

In this paper, we propose and demonstrate a solution for the rapid prototyping of low-cost soft-lithographic channel moulds for the fabrication of microfluidic channels in PDMS. The proposed open-source process employs a standard MEX 3D-printer printing with standard thermoplastic (PLA) in single line extrusions to quickly and cheaply fabricate interconnecting channel moulds which can then be connected and fused to a glass substrate through a very simple thermal bonding process.

Whole libraries of interconnecting channel moulds can be fabricated at negligible expense, thereby allowing their use for both education as well as research in resource poor settings. The major advantage of this method is the very low cost of the channel moulds as well as the speed and simplicity of the fabrication process, which requires no additional equipment, chemicals reagents [27,57] or other resources besides a heat source. This could be something as simple as a kitchen stove or Bunsen burner. In this way, our novel technique solves 2 of the 3 major

disadvantages highlighted by Kojic et.al. [19] for the use of PDMS microfluidic chips; the "non-trivial lithography method", and the "high cost of chips". The third disadvantage highlighted by Kojic et.al. [19] is the challenges in creating 3D channels in PDMS. This is an area where the authors believe the use of this 3D-printed soft-lithography approach can also help and will be subject of future work.

We also demonstrated printing of 100 μm channels, a marked improvement in standard 3D-printing channel resolution which compares favourably with the dimensions and resources used in other low-cost techniques, such as laser cutting [19], and paper-microfluidics [20,23]. The channel dimensions achieved by this method are by no means the highest resolution possible through additive manufacturing techniques, but they are the highest the authors are aware of in relation to cost-effectiveness, reliability, and simplicity. It is expected that, as the field of MEX 3D-printing further evolves, the resolutions will continue to improve.

Finally, we experimentally demonstrated the use of PDMS microfluidics fabricated using the proposed method to produce droplets and characterised their size distributions (or polydispersity). Here it was shown that our channels exhibit a polydispersity index (PDI) of <8%. Although not reaching PDI's associated with more sophisticated microfluidic chip fabrication processes (1–3%), this result is a promising starting point for further optimisation and refinement and is still reliable enough for many applications in teaching and research.

## 6. Conclusions

This simple and cost-effective technique has far-reaching impact, both for scientific research and education. The completely open-source, negligible-cost technique allows users to prototype interconnecting microfluidic channels down to 100 μm resolution in width, significantly lowering the barriers to entry for those without the resources typically needed for microfluidic fabrication. Users will be able to print their own designs with the open-source CAD add-in or request a vast library of microchannel scaffolds for the negligible cost to print them. Users will then be able to fabricate microfluidic devices from PDMS with only household equipment and no hazardous chemicals, thereby making microfluidic experimentation accessible to schools, hobbyists, and researchers no matter their resources. It is hoped that this approach will be adopted by researchers and educators around the world and will help to inspire the next generation of lab-on-a-chip developers.

Perhaps most excitingly, the simple yet robust fabrication process results in PDMS microfluidic devices that can be applied directly to glass, without requiring plasma-activation, and yet still easily support typical microfluidic testing pressures. As such, the technique could pave the way for truly affordable lab-on-a-chip healthcare diagnostic testing, performed at point-of-care, by applying the microfluidic PDMS channels directly to any cleaned glass surface, such as a mobile phone screen or car windshield.

## Supporting information

**S1 Fig. Annotated channel diagrams.** a. Cross-junction b. Droplet Generator c. Fluid Resistor d. Straight e. Y-junction.
(TIF)

**S2 Fig. Example modules prints.** (A) Failed modules caused by poor bed adhesion, due to the print head being too far from the print bed. (B) Under-extruded module caused by the print bed being too close to the print head. (C) Stringing between droplet modules. (D) Completed cross-junction modules with top showing an ideal print, and bottom over-extruded.
(EPS)

**S1 Supporting information. μ-fluid generator add-in for Autodesk Fusion.**
(ZIP)

**S2 Supporting information. Ultimaker Cura profiles.** MicroChannel01.curaprofile is for the
0.1 mm nozzle. MicroChannel04.curaprofile is for the 0.4 mm nozzle.
(ZIP)

**S1 Video. 100 μm FEM PB Droplet Count.**
(M4V)

**S2 Video. 100 μm PB Droplet Generator 240 FPS.**
(AVI)

**S3 Video. 350 μm FEM PB Droplet Generator.**
(M4V)

**S1 File. Measured channel widths for the 350 and 100 μm data sets.**
(XLSX)

## Author Contributions

**Conceptualization:** Robert Hughes.

**Formal analysis:** Harry Felton, Robert Hughes, Andrea Diaz-Gaxiola.

**Investigation:** Harry Felton, Robert Hughes, Andrea Diaz-Gaxiola.

**Methodology:** Robert Hughes, Andrea Diaz-Gaxiola.

**Project administration:** Robert Hughes.

**Software:** Harry Felton.

**Supervision:** Robert Hughes.

**Validation:** Harry Felton, Robert Hughes, Andrea Diaz-Gaxiola.

**Visualization:** Harry Felton, Robert Hughes, Andrea Diaz-Gaxiola.

**Writing – original draft:** Harry Felton, Robert Hughes, Andrea Diaz-Gaxiola.

**Writing – review & editing:** Harry Felton, Robert Hughes, Andrea Diaz-Gaxiola.

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
