## [Decision Letter · Decision Letter 0]

18 Sep 2020

PONE-D-20-23937

Negligible-cost microfluidic device fabrication using 3D-printed interconnecting channel scaffolds

PLOS ONE

Dear Dr. Hughes,

Thank you for submitting your manuscript to PLOS ONE. After careful consideration, we feel that it has merit but does not fully meet PLOS ONE’s publication criteria as it currently stands. Therefore, we invite you to submit a revised version of the manuscript that addresses the points raised during the review process.

We look forward to receiving your revised manuscript.

Kind regards,

Zhan Li, PhD

Academic Editor

PLOS ONE

Journal Requirements:

2.Thank you for stating the following in the Acknowledgments Section of your manuscript:

[The authors would like to acknowledge the training & pump-priming funding award through BristolBridge

(grant number EP/ M027546/1) under the Engineering and Physical Sciences Research Council

(EPSRC) Bridging the Gaps between the Engineering and Physical Sciences and Antimicrobial

Resistance cross-council AMR initiative, as well as pump-priming awarded by the Civil, Aerospace &

Mechanical Engineering (CAME) School, University of Bristol. Harry Felton’s work reported in this paper

has been undertaken as part of the Twinning of digital-physical models during prototyping project at the

University of Bristol, which is funded by the EPSRC [grant number EP/R032696/1]. Andrea Diaz

Gaxiola’s work has been funded by ONACYT, MEXICO (http://www.conacyt.gob.mx/).]

 [The author(s) received no specific funding for this work.]

Reviewers' comments:

Reviewer's Responses to Questions

**Comments to the Author**

1. Is the manuscript technically sound, and do the data support the conclusions?

Reviewer #1: Partly

Reviewer #2: Partly

2. Has the statistical analysis been performed appropriately and rigorously? 

Reviewer #1: Yes

Reviewer #2: Yes

3. Have the authors made all data underlying the findings in their manuscript fully available?

Reviewer #1: Yes

Reviewer #2: Yes

4. Is the manuscript presented in an intelligible fashion and written in standard English?

Reviewer #1: Yes

Reviewer #2: Yes

5. Review Comments to the Author

Reviewer #1: The authors presented a 3D printing process for the development of interconnecting microchannel scaffolds with a resolution down to 100 micron and a nearly negligible cost. The work is interesting, the 3D printed microchannel devices showed a good promise towards real-world applications, but before I can consider recommending publication in PLOS One I would suggest the authors to address the following issues:

1. To keep the manuscript written in a concise way, the authors may consider removing some of the figures in the manuscript. For example, Fig 1 is a typical process that is not necessarily placed in the introduction section of the manuscript; fig 2 describes the results of a previous work of others and therefore, the figure shall not be used in this manuscript; fig 2-5 may be put together and some of the information can be put in the supplementary.

2. I would suggest the authors not to use words like "novel" "significantly" "drastically" repeatedly.

3. The authors demonstrated several 3D printed device prototypes and discussed their printed microchannel characterisations in a detailed manner. In this perspective, the readers can easily evaluate the 3D printing results. However, it is not possible to determine whether the printed devices perform well as there lacked control experiments of microchannel devices fabricated via other means in this work or there are no references to the studies of others to allow the readers to compare the device performance. I would suggest the authors to consider put more efforts on the discussion of the device performance.

Reviewer #2: The paper presented a microfluidic device fabrication paradigm based on 3D printing technology. The modularized design and manufacturing of the channel scaffolds accelerated the realization of microfluidic systems with low-cost. Through the authors giving thorough descriptions of every detail of the printing settings, this manuscript's major issue is the lack of sound analysis on explaining the reason for choosing fabrication parameters. Some other technical comments:

1. The descriptions on the 3d printing technology, 3d printer, and the slicing software in this work is trivial since the layer manufacturing has already been employed in the microfluidic and microelectronics fields with a lot of achievements, referring to ref. [1]. The manuscript should be shortened with a concise introduction of the settings related to mold design and fabrication.

2. In an FDM, process planning, which includes slicing and print path planning, is vital in determining print quality. Instead of listing all the printer settings, to improve the printing quality (section 2.2.2), the authors are suggested to further investigating the key parameters (like slicing height, line width, infill pattern, et al.) that affect the material deposition. Many works related to the print quality improvement of FDM has been overlooked, such as ref. [2,3]. Furthermore, the qualitative evaluation results listed in Table 2 seems meaningless for other practitioners or other print parts. The quantified analysis of the reason for unsuccessful printing should be made.

3. What are the advantages of this work over other 3d printed microfluidic devices, such as ref? [4]. To the best of my understanding, the device fabricated by the method of this work is simply two dimensional (the height is constant). Can the method be extended to a more complex device with variable height? Moreover, the authors may show some comparisons with others’ works.

The 3d printed microfluidic device is an exciting topic, and the low-cost one is desperately needed in the academic sector. The author presented a complete process in realizing the fast fabrication of the device. However, the current manuscript is more like an introduction to instructing practitioners to design and manufacture the microfluidic device step-by-step. The manuscript has too much trivial information related to the print settings, which could be shortened, while lacks sufficient explanations on the key factors that affect the deposition accuracy or print surface quality. More quantified analysis related to the relationship between print parameters and the accuracy of the fabricated part should be added. The effectiveness of more demonstrations of the device can also be validated.

Based on the above concerns, I suggest a major revision of this paper.

Reference:

[1] R. D. Sochol et al., “3D printed microfluidics and microelectronics,” Microelectronic Engineering, vol. 189. pp. 52–68, 2018.

[2] A. Dey and N. Yodo, “A systematic survey of FDM process parameter optimization and their influence on part characteristics,” J. Manuf. Mater. Process., vol. 3, no. 3, p. 64, 2019.

[3] G. Q. Jin, W. D. Li, and L. Gao, “An adaptive process planning approach of rapid prototyping and manufacturing,” Robot. Comput. Integr. Manuf., vol. 29, no. 1, pp. 23–38, 2013.

[4] S. Tsuda et al., “Customizable 3D printed ‘Plug and Play’ millifluidic devices for programmable fluidics,” PLoS One, vol. 10, no. 11, pp. 1–13, 2015.

6. PLOS authors have the option to publish the peer review history of their article (what does this mean?). If published, this will include your full peer review and any attached files.

Reviewer #1: No

Reviewer #2: No

---

## [Author Response · Author response to Decision Letter 0]

12 Nov 2020

Please see attached response to reviewers comments document which is highlighted for easy reading.

Reviewer #1: 

The authors presented a 3D printing process for the development of interconnecting microchannel scaffolds with a resolution down to 100 micron and a nearly negligible cost. The work is interesting, the 3D printed microchannel devices showed a good promise towards real-world applications, but before I can consider recommending publication in PLOS One I would suggest the authors to address the following issues:

The authors would like to thank Reviewer 1 for their helpful comments and suggestions.

1. To keep the manuscript written in a concise way, the authors may consider removing some of the figures in the manuscript…

…For example, Fig 1 is a typical process that is not necessarily placed in the introduction section of the manuscript; 

Figure 1 has been removed and all references to it deleted. The authors now reference suitable literature for the process.

…fig 2 describes the results of a previous work of others and therefore, the figure shall not be used in this manuscript; 

The authors have noticed that the wording in the manuscript relating to Figure 2 (now Fig.1) may have caused confusion. All of the images in Figure 2 were produced by the authors themselves and corroborate the work of others as referenced in the manuscript. The figure itself is important for successfully communicating these motivations and correctly framing the work so has been kept. However, the authors have made clearer in the manuscript that the work is original and not from the work of others.

…fig 2-5 may be put together and some of the information can be put in the supplementary.

Thank you for your recommendation. The authors have removed Figure 5 and now include it in the supplementary material (see S.8). We have also greatly simplified the description of print failures in the main manuscript. However, the authors feel that the remaining figures (originally figures 2-4, now 1-3) are individually important and more informative when separated in their respective positions, as they each discuss different aspects of the work. The authors therefore feel that combining them would result in an overly large and complicated figure that would confuse more than enhance the reading of the manuscript. This approach appears consistent with other PLOS One papers. 

2. I would suggest the authors not to use words like "novel" "significantly" "drastically" repeatedly.

Thank you for your suggestion. We acknowledge the overuse of certain words and have altered the manuscript to eliminate excessive occurrences of “novel” and “significantly”. On inspection, there were no uses of the word “drastically” in the original manuscript.

3. The authors demonstrated several 3D printed device prototypes and discussed their printed microchannel characterisations in a detailed manner. In this perspective, the readers can easily evaluate the 3D printing results. However, it is not possible to determine whether the printed devices perform well as there lacked control experiments of microchannel devices fabricated via other means in this work or there are no references to the studies of others to allow the readers to compare the device performance. I would suggest the authors to consider put more efforts on the discussion of the device performance.

This is a very helpful point. Thank you. Due to the current restrictions in place due to COVID, additional control experiments have not been possible at this time. The authors have taken the opportunity to further examine and compare the devices to the performance of similar devices fabricated using other methods obtained from the literature and have discussed their performance more thoroughly, specifically in relation to the polydispersity of droplets from droplet generators. We have also now included a discussion to highlight the key advantages and compared to other works. The authors believe that this satisfies the concerns and helps to further validate the approach. 

Reviewer #2: 

The paper presented a microfluidic device fabrication paradigm based on 3D printing technology. The modularized design and manufacturing of the channel scaffolds accelerated the realization of microfluidic systems with low-cost. Through the authors giving thorough descriptions of every detail of the printing settings, this manuscript's major issue is the lack of sound analysis on explaining the reason for choosing fabrication parameters. Some other technical comments:

The authors would like to thank Reviewer 2 for their valuable suggestions and comments. Please see more detailed comments below regarding the analysis of the parameters chosen.

1. The descriptions on the 3d printing technology, 3d printer, and the slicing software in this work is trivial since the layer manufacturing has already been employed in the microfluidic and microelectronics fields with a lot of achievements, referring to ref. [1]. The manuscript should be shortened with a concise introduction of the settings related to mold design and fabrication.

The introduction to this section has now been made more concise, and discusses other work, however, the authors feel it is important to specify the printer being used, along with the material, as this can have a significant effect on the results of the testing. Changing to a printer with a dual-geared feed, for example, would improve the control over material feed rate and would therefore change the statistics around print success. However, testing has now been completed with alternative brands of PLA – with minor modifications to print settings – and this has been made clear in the text.

2. In an FDM, process planning, which includes slicing and print path planning, is vital in determining print quality. Instead of listing all the printer settings, to improve the printing quality (section 2.2.2), the authors are suggested to further investigating the key parameters (like slicing height, line width, infill pattern, et al.) that affect the material deposition. Many works related to the print quality improvement of FDM has been overlooked, such as ref. [2,3]. 

The authors are not sure what the reviewer is referencing here. The print settings, in this instance at least, are the same as print parameters (slicing height – or layer height – and line width are both specified in Table 1). Infill pattern is not appropriate for discussion within the paper as the prints are either single line width or shell prints only. Having read the references provided, they discuss the effect of print parameters on parts of a much larger scale – several mm in each direction rather than µm - and thus are not directly applicable to this work.

We would like to thank you for raising the points surrounding print path planning as we have omitted this from the original paper. As such we have added a paragraph discussing why we have not manually – or with an original piece of code – planned out print paths, instead allowing Cura to do this for us. In brief, our reasons for doing this were to allow non-expert users (the target for this technology in many instances) to quickly adopt the process. We hope that these changes satisfy the reviewers concerns.

…Furthermore, the qualitative evaluation results listed in Table 2 seems meaningless for other practitioners or other print parts. The quantified analysis of the reason for unsuccessful printing should be made.

Once again, thank you to the reviewer for their comments. Based on their feedback the authors have added a section analysing the size distribution of the 3D printed moulds to define the groupings more clearly as a replacement for Table 2, providing a quantitative distribution instead of a qualitative grouping. In short, 500 scaffolds were printed over the two channel sizes and measured, producing lognormal distributions for the channel widths produced. The raw sizing data is provided in the supplementary data (S.7).

The failure mechanisms have also been quantitively described using the same data set as the distributions now use; albeit with the caveat that this is highly variable depending on the print bed levelling and filament feed control. Clearer definitions of the context we have used the terms “Print Success” and “Useable Print” in are also provided. This work has been presented for the 350 µm and 100 µm channels.

With regards to the applicability to other print parts, the methods, settings, and statistics presented in the paper directly relate to the designs discussed therein. However, the authors expect future work to consider further designs and, through making the methods publicly available, fully support others investigating the methods applicability to other designs.

3. What are the advantages of this work over other 3d printed microfluidic devices, such as ref? [4]. To the best of my understanding, the device fabricated by the method of this work is simply two dimensional (the height is constant). Can the method be extended to a more complex device with variable height? Moreover, the authors may show some comparisons with others’ works.

Thank you for your questions. The devices fabricated using this method are indeed 2D at the moment, as are many microfluidic devices being used at present, however, the authors believe that more complex 3D designs would indeed be possible with some alterations to the fabrication process. This will be the focus of subsequent work by the authors. 

The authors see the major advantages this process has over other 3D-printed devices as being the very rapid printing and fabrication time (~<5mins), the very low cost per device, the reduction in channel resolution (100um) for FDM prints, and the method reliability (in part due to the above 2 points and the ability to print sections independently so whole prints do not have to be thrown out if errors occur). In addition, this method uses minimal additional resources (i.e. oxygen plasma treatment or chemicals such a dichloromethane used in the reference mentioned by the reviewer) to fabricate the final devices, besides a method of heating, which are readily available. 

To clarify the above, the authors have now included a discussion section to summarise the advantages of the work as well as compare to other approaches. We have also (as recommended by Reviewer 1) made a critical assessment and comparison of the polydispersity of droplets produced by the droplet generation devices fabricated using this method, and compared them to others’ works in order to evaluate the efficiency of the technique. We hope and believe that these additions will satisfy the reviewers concerns.

The 3d printed microfluidic device is an exciting topic, and the low-cost one is desperately needed in the academic sector. The author presented a complete process in realizing the fast fabrication of the device. However, the current manuscript is more like an introduction to instructing practitioners to design and manufacture the microfluidic device step-by-step. The manuscript has too much trivial information related to the print settings, which could be shortened, while lacks sufficient explanations on the key factors that affect the deposition accuracy or print surface quality. More quantified analysis related to the relationship between print parameters and the accuracy of the fabricated part should be added…

Thank you for your feedback. As discussed in relation to comments by Reviewer 1 and your previous comment, the authors have re-edited the sections relating to print settings to eliminate trivial or overly verbose explanations in an effort to make the manuscript more concise and direct. However, this paper is meant to be instructive for non-expert users, and as such we believe that the information presented to allow a practitioner to undertake each step is important.

In addition to this, and as previously stated, the machine used will have a considerable effect on the quality of the printed artefacts. As such, a detailed quantified analysis of the effect of print settings for this specific printer did not seem as valuable as detailing the performance of the fabricated channels – something we have now done in more detail – for this introductory work. It is also important to understand that there are ~500 settings that can be changed in Cura, many of which will affect the parts being printed. To thoroughly discuss the important settings would require a substantial investigation with a considerably different focus. As the paper originally presented, we have listed some of the key print settings/parameters with a brief explanation to provide context to the results included. We hope this is satisfactory.

However, based on your feedback, we have added a sizing analysis for the settings used to demonstrate more clearly what is being achieved and act as a line in the sand for anyone who may replicate the procedure. Future papers are planned that aim to investigate the effect individual settings/groups of settings have on the print quality and success in more depth.

With regards to print surface quality, this is not a concern using this method (as shown by Fig.5) due to the smoothing effect of the thermal bonding process. As such the channel surface quality and repeatability of channel dimensions are both improved (Fig.7). 

…The effectiveness of more demonstrations of the device can also be validated…

The authors have included additional droplet sizing evaluation, such that the devices’ performance has now been further analysed and validated against other literature resources with similar purpose, such as droplet generators, that have been fabricated through other methods.

Based on the above concerns, I suggest a major revision of this paper.

Additional Changes:

In addition to the above changes, the authors have also made a number of minor alterations to certain wording and phrasing in order to improve clarity and make the manuscript more concise.

The authors have also included additional droplet generation data and microscope images in Fig.9 to support the analysis of droplet polydispersity in the text.

Thank you again for you time and consideration of our manuscript. We appreciate all of the suggestions and believe these have greatly improved the quality of the paper.

---

## [Decision Letter · Decision Letter 1]

18 Dec 2020

PONE-D-20-23937R1

Negligible-cost microfluidic device fabrication using 3D-printed interconnecting channel scaffolds

PLOS ONE

Dear Dr. Hughes,

Thank you for submitting your manuscript to PLOS ONE. After careful consideration, we feel that it has merit but does not fully meet PLOS ONE’s publication criteria as it currently stands. Therefore, we invite you to submit a revised version of the manuscript that addresses the points raised during the review process.

We look forward to receiving your revised manuscript.

Kind regards,

Zhan Li, PhD

Academic Editor

PLOS ONE

Reviewers' comments:

Reviewer's Responses to Questions

**Comments to the Author**

1. If the authors have adequately addressed your comments raised in a previous round of review and you feel that this manuscript is now acceptable for publication, you may indicate that here to bypass the “Comments to the Author” section, enter your conflict of interest statement in the “Confidential to Editor” section, and submit your "Accept" recommendation.

Reviewer #1: All comments have been addressed

Reviewer #2: All comments have been addressed

2. Is the manuscript technically sound, and do the data support the conclusions?

Reviewer #1: Yes

Reviewer #2: Yes

3. Has the statistical analysis been performed appropriately and rigorously? 

Reviewer #1: Yes

Reviewer #2: I Don't Know

4. Have the authors made all data underlying the findings in their manuscript fully available?

Reviewer #1: Yes

Reviewer #2: Yes

5. Is the manuscript presented in an intelligible fashion and written in standard English?

Reviewer #1: Yes

Reviewer #2: Yes

6. Review Comments to the Author

Reviewer #1: Thank the authors for the revision of this manuscript. The authors have addressed my previous querries and concerns. I therefore recommand publication in Plos One.

Reviewer #2: The authors have made significant improvements in the revised version of the manuscript. Most of my concerns have alleviated. The 3d-printing-based microfluidic device fabrication method introduced in the paper substantially lowers the barrier for undergraduate students and other industry practitioners with low cost and easy-to-implement processes. The authors elaborate on all the 3d modeling, printer setting, and fabrication parameters. Before considering for publication, some minor issues are expected to be resolved:

1. Page 2, Figure 1, I cannot find figure 1(g), which is introduced in the caption of Figure 1.

2. Table 1, the unit of the build plate temperature is wrong.

3. Make sure all the literature is correctly referenced (e.g., Line 213).

7. PLOS authors have the option to publish the peer review history of their article (what does this mean?). If published, this will include your full peer review and any attached files.

Reviewer #1: No

Reviewer #2: No

---

## [Author Response · Author response to Decision Letter 1]

21 Dec 2020

Reviewer #1: Thank the authors for the revision of this manuscript. The authors have addressed my previous querries and concerns. I therefore recommand publication in Plos One.

Thank you very much for your feedback and your constructive comments. They were greatly appreciated.

Reviewer #2: The authors have made significant improvements in the revised version of the manuscript. Most of my concerns have alleviated. The 3d-printing-based microfluidic device fabrication method introduced in the paper substantially lowers the barrier for undergraduate students and other industry practitioners with low cost and easy-to-implement processes. The authors elaborate on all the 3d modeling, printer setting, and fabrication parameters. Before considering for publication, some minor issues are expected to be resolved:

We would like to thank the reviewer for their comments and attention to detail. We have corrected these issues.

1. Page 2, Figure 1, I cannot find figure 1(g), which is introduced in the caption of Figure 1.

Thank you very much for noticing this. This has been corrected.

2. Table 1, the unit of the build plate temperature is wrong.

Again, we thank you for pointing this out. This has been corrected.

3. Make sure all the literature is correctly referenced (e.g., Line 213).

Thank you. We have corrected this issue.

---

## [Editor Report · Decision Letter 2]

23 Dec 2020

PONE-D-20-23937R2

Negligible-cost microfluidic device fabrication using 3D-printed interconnecting channel scaffolds

PLOS ONE

Dear Dr. Hughes,

Thank you for submitting your manuscript to PLOS ONE. After careful consideration, we feel that it has merit but does not fully meet PLOS ONE’s publication criteria as it currently stands. Therefore, we invite you to submit a revised version of the manuscript that addresses the points raised during the review process.

We look forward to receiving your revised manuscript.

Kind regards,

Zhan Li, PhD

Academic Editor

PLOS ONE

Additional Editor Comments (if provided):

There are still some errors and typos as pointed out by the reviewer, which are not yet corrected, e.g., "...Images e-g) show characteristic ridges of 101 channels manufactured using SLA moulds printed at an angle to the substrate plane to improve print resolution..." in the caption of Fig. 1.

---

## [Author Response · Author response to Decision Letter 2]

23 Dec 2020

Reviewer #1: Thank the authors for the revision of this manuscript. The authors have addressed my previous querries and concerns. I therefore recommand publication in Plos One.

Thank you very much for your feedback and your constructive comments. They were greatly appreciated.

Reviewer #2: The authors have made significant improvements in the revised version of the manuscript. Most of my concerns have alleviated. The 3d-printing-based microfluidic device fabrication method introduced in the paper substantially lowers the barrier for undergraduate students and other industry practitioners with low cost and easy-to-implement processes. The authors elaborate on all the 3d modeling, printer setting, and fabrication parameters. Before considering for publication, some minor issues are expected to be resolved:

We would like to thank the reviewer for their comments and attention to detail. We have corrected these issues.

1. Page 2, Figure 1, I cannot find figure 1(g), which is introduced in the caption of Figure 1.

Thank you very much for noticing this, and to the editor for highlighting this again. This was an issue with the figure itself not the caption. We have therefore uploaded a corrected figure with the correct lettering to remedy this issue.

2. Table 1, the unit of the build plate temperature is wrong.

Again, we thank you for pointing this out. This has been corrected.

3. Make sure all the literature is correctly referenced (e.g., Line 213).

Thank you. We have corrected this issue.

The authors have also conducted a final proof-read the manuscript as recommended by the editor, and have corrected a number of minor errors and typos shown in the tracked changes version of the manuscript. We therefore believe we have corrected all outstanding issues raised by the reviewers and the editor.

---

## [Editor Report · Decision Letter 3]

26 Dec 2020

Negligible-cost microfluidic device fabrication using 3D-printed interconnecting channel scaffolds

PONE-D-20-23937R3

Dear Dr. Hughes,

We’re pleased to inform you that your manuscript has been judged scientifically suitable for publication and will be formally accepted for publication once it meets all outstanding technical requirements.

Kind regards,

Zhan Li, PhD

Academic Editor

PLOS ONE
---

## [Editor Report · Acceptance letter]

18 Jan 2021

PONE-D-20-23937R3 

Negligible-cost microfluidic device fabrication using 3D-printed interconnecting channel scaffolds 

Dear Dr. Hughes:

I'm pleased to inform you that your manuscript has been deemed suitable for publication in PLOS ONE. Congratulations! Your manuscript is now with our production department. 

Kind regards, 

on behalf of

Dr. Zhan Li 

Academic Editor

PLOS ONE